# Efficient and Selective Adsorption of Gold Ions from Wastewater with Polyaniline Modified by Trimethyl Phosphate: Adsorption Mechanism and Application

**DOI:** 10.3390/polym11040652

**Published:** 2019-04-09

**Authors:** Chen Wang, Jiling Zhao, Shixing Wang, Libo Zhang, Bing Zhang

**Affiliations:** 1National Local Joint Laboratory of Engineering Application of Microwave Energy and Equipment Technology, Kunming 650093, China; wc932608335@163.com (C.W.); 15687884157@163.com (J.Z.); 15198750002@163.com (B.Z.); 2State Key Laboratory of Complex Nonferrous Metal Resources Clean Utilization (Kunming University of Science and Technology), Kunming 650093, China; 3Faculty of Metallurgical and Energy Engineering, Kunming University of Science and Technology, Kunming 650093, China

**Keywords:** Au(III), polymer, wastewater, adsorption, mechanism

## Abstract

The selective recovery of gold from wastewater is necessary because it is widely used in various fields. In this study, a new polymeric adsorbent (TP-AFC) was prepared by modifying polyaniline with trimethyl phosphate for the selective recovery of gold from wastewater. Bath experiments were carried out to explore the adsorption capacity and mechanism. The optimum pH of adsorption is 4. The adsorption equilibrium is reached at 840 min. The maximum adsorption capacity is 881 mg/g and the adsorption was a spontaneous endothermic process. The adsorption process fitted well with pseudo second-order kinetic and the Langmuir-models. The single-layer chemisorption governed the adsorption process. In addition, the application in wastewater indicated that the interfering ions had no effect on the adsorption of gold ions. TP-AFC has good selectivity. The interaction mechanism was mainly ion exchange and complexation. In general, TP-AFC was successfully prepared and has an excellent future in practical application.

## 1. Introduction

In modern industry, precious metals are widely used in various fields [1]. As one of the most important precious metals, gold was wildly used in electronics catalysis, medicine and the jewelry industry [2,3,4,5]. As far as capacity is concerned, the supply of gold is far from meeting the growing demand. Gold ions can also trigger bioaccumulation problems when they were emitted into the aqueous solution. Therefore, it is very important to recover gold from wastewater. 

Recently, various approaches have been developed to recover gold from wastewater, including electrolysis, solvent extraction, membrane filtration and adsorption [6,7,8,9,10]. In the above methods, adsorption is a valid method with the advantage of low cost, high efficiency and easy synthesis [11]. A variety of adsorbents have been used to remove metal ions from aqueous solutions, including resin, zeolites, nanomaterials and polymer [12,13,14,15]. Among them, functional polymers have been extensively applied to the adsorption field, such as polyaniline [16,17]. Usually, the polymers were typically used as inactive carrier materials to increase the particle size and effective specific surface area [18,19,20]. However, the total volume and weight of adsorbents are greatly increased due to adding the carrier materials. These factors decrease the adsorption capacity. Hence, on the premise of ensuring adsorption capacity and selectivity, the choice of carrier material is also very important for the polymer adsorbent. 

In general, for rare and expensive metal ion adsorbents, phosphorus-containing functional groups are used to embellish them, including P–O, P=O, P–C and etc. As a cheap industrial product, trimethyl phosphate is rich in phosphorus functional groups (P–O, P=O) [21]. However, there are few reports on the modification of polymers with trimethyl phosphate.

In this research, polyaniline-based adsorbent (TP-AFC) was successfully synthesized by grafting trimethyl phosphate. The adsorbent was characterized by Fourier transform infrared spectroscopy (FT-IR), X-ray photoelectron spectroscopy (XPS), Scanning Electron Microscope (SEM), and Energy Dispersive Spectrometer (EDS). In batch adsorption experiments, several factors were investigated e.g. pH, initial concentration, reaction time, and temperature. The practical application in wastewater with coexisting ions was examined. The adsorption mechanism was investigated by batch experiments, adsorption isothermal, and kinetic models.

## 2. Materials and Methods

### 2.1. Materials

Aniline, isopropanol, formaldehyde, acetic acid, and trimethyl phosphate were purchased from Aladdin Chemical Co., Ltd. (Shanghai, China). The reagents were analytical grade and were not further processed. Gold ion standard solution and mixed ion standard solution were provided by Tianjin Zhiyuan Co., Ltd. (Tianjin, China). Hydrochloric acid and sodium hydroxide were used to adjust the pH of the solution.

### 2.2. Synthesis of Adsorbent

TP-AFC is synthesized according to Scheme 1. Polyaniline was prepared according to the literature [22]. Firstly, 25 mL isopropanol, 0.03 mol of aniline and 3 mL concentrated hydrochloric acid (12 mol/L) were added to beaker A and kept on an ice bath. 5 mL formaldehyde (37%) and 25 mL isopropanol were sequentially added to beaker B. Finally, the solution in beaker B was added to the previously cooled beaker A and stirred vigorously to obtain a homogeneous solution. After reacting at 273–278 K for 25 min, it was slowly warmed to room temperature and polyaniline was obtained.

3 g polyaniline, 0.06 mol trimethyl phosphate, 2 mL acetic acid and 10 mL formaldehyde were sequentially added to a three-necked flask and stirred at 358 K for 3 h. The final product was obtained after centrifugation. This product was named TP-AFC.

### 2.3. Batch Experiments

Batch adsorption experiments were carried out to study the influence of pH, initial concentration, and reaction time. 10 mg TP-AFC was firstly added to a 20 mL solution. The solution was oscillated at 298 K and 250 rpm. In addition, the effect of temperature on the absorption capacity was studied at 298 K, 308 K and 318 K. In order to investigate the reusability of the adsorbent, a repetitive experiment was conducted. A thiourea hydrochloride desorbent was used. In order to study the practical application, 10 mg TP-AFC was added to 20 mL laboratory wastewater and oscillated at 298 K and 250 rpm. In all batch experiments, the supernatant was collected after centrifugation at 9000 rpm for 10 minutes and the remaining ion concentration was measured using inductively coupled plasma optical emission spectrometer. The removal rate of gold ions (R) and the equilibrium adsorption capacity ($q_{x}$) were calculated by the following equation [23].
(1)$$R=\frac{\left(C_{y}-C_{r}\right)}{C_{y}}\times 100\%$$
(2)$$q_{x}=\frac{\left(C_{y}-C_{r}\right)}{m}V$$
where $C_{y}$ and $C_{r}$ are the initial concentration and residual concentration of gold ions, respectively. V (mL) and m (mg) are the volume of the gold ion solution and the weight of TP-AFC, respectively.

### 2.4. Characterization

Fourier transform infrared spectroscopy (FT-IR) was measured by Nicolet iS50 FT-IR spectrophotometer (Thermo Nicolet, Waltham, MA, USA) between 400 cm^−1^ and 4000 cm^−1^ with a resolution of 4 cm^−1^. Scanning-Electron-Microscope (SEM, Phenom pro X, Netherlands.) was used to characterize the size and morphology of polyaniline and TP-AFC. The surface state of the adsorbent during adsorption was characterized by XPS (Thermo Scientific Co., 1486.6 eV monochromated Al K-alpha radiation source, Chanhassen, MN, USA). The concentration of metal ions was detected by inductively coupled plasma optical emission spectrometer (ICP-OES, LEEMAN prodigy 7, Hudson, Wenthworth Drive, NH, USA).

## 3. Results and Discussion

### 3.1. Characterization of TP-AFC

The FT-IR spectra of polyaniline and TP-AFC are shown in Figure 1. The peaks at 820 cm^−1^, 1667 cm^−1^, 1596 cm^−1^ and 3414 cm^−1^ represented the CH, C-N, N-H and O-H bonds, respectively. TP-AFC showed two new peaks compared to polyaniline at 943 cm^−1^ and 1051 cm^−1^. They represented the P-O and P=O bonds, respectively. This indicated that the trimethyl phosphate was successfully grafted.

The SEM images of polyaniline and TP-AFC are shown in Figure 2. It can be found that the morphology of the adsorbent does not substantially change after being grafted. This showed that the adsorbent had a stable structure. The modification rate of the amine can be calculated by the data obtained by SEM-EDS and the structural formula of polymer. One nitrogen atom is grafted with two phosphorus atoms. The calculated modification rate of amine is 20.6%.

### 3.2. Effect of pH 

Solution pH has an important influence on the removal ability of metal ions. The effect of pH from 2 to 11 was studied when the concentration of gold ion was 200 mg/L. Figure 3a showed that the adsorption capacity varies greatly with the change in pH. Obviously, TP-AFC shows the excellent adsorption ability in the pH range of 2 to 6. The maximum removal rate of gold ions by TP-AFC was occurred at pH 4. In contrast, the good removal rate of gold ions by polyaniline was occurred in the pH range of 2 to 4. When the pH is too high, the hydroxide(OH^−^) in the solution competes with the adsorbent, resulting in a sharp drop in the adsorbing capacity. As the pH of the solution changes, gold ions mainly exist in the following forms:$$\begin{array}{l} \mathrm{Au}{\left(\mathrm{OH}\right)}_{4}+H^{+}+{\mathrm{Cl}}^{-}⇌\mathrm{AuCl}{\left(\mathrm{OH}\right)}_{3}^{-}+H_{2}O\\ \mathrm{AuCl}{\left(\mathrm{OH}\right)}_{3}^{-}+H^{+}+{\mathrm{Cl}}^{-}⇌{\mathrm{AuCl}}_{2}{\left(\mathrm{OH}\right)}_{2}^{-}+H_{2}O\\ {\mathrm{AuCl}}_{2}{\left(\mathrm{OH}\right)}_{2}^{-}+H^{+}+{\mathrm{Cl}}^{-}⇌{\mathrm{AuCl}}_{3}{\left(\mathrm{OH}\right)}^{-}+H_{2}O\\ {\mathrm{AuCl}}_{3}{\left(\mathrm{OH}\right)}^{-}+H^{+}+{\mathrm{Cl}}^{-}⇌{\mathrm{AuCl}}_{4}^{-}+H_{2}O \end{array}$$

The net total surface charge of TP-AFC can be represented by the zero potential (pH_PZC_). The value of pH_PZC_ charge can be determined by the method described in the literature [24]. The pH_PZC_ of TP-AFC was 2.8. In order to study the adsorption mechanism, the zeta potential of TP-AFC and gold solution were also determined. As shown in Figure 3b, the isoelectric point of TP-AFC was 4.6, indicating that the potential was positive when pH was below 4.6. The gold solution showed a negative charge in the range of pH 2 to pH 4. Hence, there has an electrostatic effect between TP-AFC and gold.

### 3.3. Effect of Reaction Time

The effect of the reaction time on adsorption capacity was studied in order to explore the adsorption mechanism. 10 mg TP-AFC was added to 20 mL gold solutions (pH 4) and oscillated at 298 K with 250 rpm. The time was changed from 10 to 1440 min. As can be seen from Figure 4a, the removal rate increased from 14% to 95% as time increased from 10 to 840 min, because many adsorption sites can rapidly combine with gold ions during the adsorption stage. As time goes by, the adsorption sites are few. Therefore, the removal rate did not change significantly after 840 min. Therefore, 840 min is the optimal reaction time.

The adsorption kinetics was studied by pseudo-first-order kinetic model, pseudo-second-order kinetic model, and intraparticle kinetic model. The pseudo-first-order kinetic model assumes that the adsorption rate varies with the number of non-adsorbed sites on the surface of the adsorbent. Equation (3) is used to describe the relationship between them [25]:(3)$$ln\left(q_{b}-q_{t}\right)=lnq_{b}-k_{1}t$$
where $q_{b}$ and $q_{t}$ are the adsorption capacities at equilibrium time and at time t, respectively. $k_{1}$ is the rate constant of the Pseudo-first-order kinetic model. It can be seen from Figure 4b and Table 1 that the value of the correlation coefficient is lower, and the theoretical equilibrium adsorption capacity (292.5 mg/g) is much lower than the actual equilibrium adsorption capacity (399.6 mg/g). Therefore, the adsorption process does not conform to the pseudo-first-order kinetic model.

The pseudo-second-order kinetic model is based on chemisorption and is affected by the mass balance equation and the second-order rate derivative. Equation (4) is used to describe the pseudo-second-order kinetic model [26]:(4)$$\frac{t}{q_{t}}=\frac{1}{k_{2}q_{b}^{2}}+\frac{t}{q_{b}}$$
where $k_{2}$ is the rate constant of the pseudo-second-order kinetic model. It can be seen from Figure 4c and Table 1 that the correlation coefficient of the pseudo-second-order kinetic model (0.999) is much closer to 1 than that of the pseudo-first-order kinetic model (0.910). In addition, the theoretical equilibrium adsorption capacity (420 mg/g) calculated by the pseudo-second-order kinetic model is close to the actual equilibrium adsorption capacity (399.6 mg/g) and the △q value of the pseudo-second-order kinetic model is smaller than that of the pseudo-first-order kinetic model. Therefore, the pseudo-second-order kinetic model described the adsorption process and the whole process was chemsorption.

The intraparticle diffusion model can further explain the adsorption process. Equation (5) is used to describe the intraparticle diffusion model [27]:(5)$$q_{b}=k_{3}t^{1/2}+C$$
where $k_{3}$ and C are the constants of the rate and the intraparticle diffusion model, respectively. As can be seen from Figure 4d and Table 2, the intraparticle diffusion model is a complex linear relationship that can be divided into three stages. The first stage was a rapid adsorption process. This stage was mainly the diffusion of gold ions from the solution to the surface of the adsorbent. The second stage was a slow adsorption process. Gold ions diffused from the surface of the adsorbent to the interior of the adsorbent. At the last stage, there was no significant change in the amount of adsorption. At this stage, the gold ions diffused slowly and the adsorption sites had reached saturation.

### 3.4. Effect of Initial Concentration 

Isothermal adsorption experiments are very important to study the adsorption mechanism. 10 mg TP-AFC was added to 20 mL gold solution with different concentrations (285 mg/L, 380 mg/L, 475 mg/L, 570 mg/L 665 mg/L and 760 mg/L), respectively. It can be seen from Figure 5a, the adsorption capacity of gold ions increased with the increasing of initial concentration. A high concentration of gold solution contributed to the binding of the adsorption site to the gold ions. The maximal adsorption capacity of TP-AFC for gold ions is 881 mg/g. The adsorption test of polyaniline was also carried out under the same conditions. Figure 5a also shows that the maximal adsorption capacity of polyaniline for gold ions is 599 mg/g, which is lower than that of TP-AFC. 

In order to further study the adsorbing mechanism, three isothermal models were used to describe the adsorption process, including Langmuir, Freundlich and Temkin. Langmuir model describes single layer adsorption with no lateral effects or spatial barriers. Langmuir model can be expressed by Equation (6) [28]:(6)$$q_{c}=\frac{q_{m}\cdot K_{L}\cdot C_{r}}{1+K_{L}\cdot C_{r}}$$
where $q_{c}$ and $C_{r}$ represent the equilibrium adsorption capacity and the remaining concentration, respectively. $K_{L}$ is the constant of Langmuir model and $q_{m}$ is the maximum theoretical adsorption capacity.

The Freundlich model describes that the adsorption process occurs on a heterogeneous surface. The Freundlich model can be expressed by Equation (7) [29]:(7)$$q_{c}=K_{F}\cdot C_{r}^{\frac{1}{n_{1}}}$$
where $K_{F}$ represents the Freundlich constant and *n*_1_ is the adsorption capacity index.

The Temkin model describes that the adsorption heat continually decreases as the adsorbent and adsorbate interact in different adsorbent layers. This model is based on chemisorptions. It can be expressed by Equation (8) [30]:(8)$$q_{c}=\frac{RTln\left(K_{T}C_{r}\right)}{\beta }$$
where $K_{T}$ and β is the constant of the Temkin model. R and T are the constant of universal gas and temperature in Kelvin, respectively.

Figure 5 b–d is the fitting curves of the three models. The relevant parameters of the models are listed in Table 3. The correlation constant (1/n) in the Freundlich model is in the range of 0 to 1, indicating that adsorption is advantageous. The Temkin model isotherm provided additional information on the adsorbent–adsorbate binding interaction. The adsorption heat from the Temkin model indicated a strong interaction between the reactive group and the Au(III) ion. The correlation coefficient is R^2^_L_>R^2^_T_>R^2^_F_. At the same time, the theoretical maximum adsorption capacity was 883.05 mg/g, which is close to the actual adsorption capacity (881mg/g). △q value of Langmuir (2.01mg/g) is smaller than that of Freundlich (275.77mg/g). Therefore, the Langmuir model well described the adsorption process.

The separation factor K_s_ can be obtained from the Langmuir model. The K_s_ value can be used to judge the superiority of the adsorption process. When K_s_ = 1, the adsorption is reversible. When 0 < K_s_ < 1, adsorption is advantageous and linear. When K_s_ > 1, adsorption is disadvantageous. The value of K_s_ can be expressed by Equation (9):(9)$$K_{s}=\frac{1}{1+K_{L}\cdot C_{r}}$$

Table 4 shows that the K_s_ values of each Au(III) initial concentration were between 0.0015 to 0.004, indicating that the adsorption process was favorable. 

In order to study the difference between TP-AFC and other adsorbents, it was compared with various adsorbents, including adsorption capacity and pH. As shown in Table 5, TP-AFC exhibits a higher adsorption capacity. 

### 3.5. Thermodynamic Experiment

The effect of temperature (293 K, 298 K and 303 K) on the adsorption capacity was investigated under pH 4 conditions. The increase of temperature has a positive effect on the adsorption of Au(III) by TP-AFC. Gibbs free energy ($\Delta G_{T}$), enthalpy ($\Delta H_{T}$), and entropy ($\Delta S_{T}$) can be obtained from Equations (10) and (11):(10)$$\ln K_{Q}=ln\frac{q_{c}}{C_{r}}=\frac{\Delta S_{T}}{R}-\frac{\Delta H_{T}}{RT}$$
(11)$$\Delta G_{T}=-RTln\frac{q_{c}}{C_{r}}$$

As shown in Figure 6 and Table 6, the value of $\Delta H_{T}$ and $\Delta S_{T}$ were positive, indicating that the adsorption is an endothermic process. The value of $\Delta G_{T}$ was negative, indicating that the adsorption is spontaneous.

### 3.6. Reusability and Desorption Studies of TP-AFC

Recyclability of materials is important for their potential applications in field of water purification. In order to investigate the regenerative capacity of TP-AFC, a repetitive experiment was conducted. 30 mg TP-AFC was added to 60 mL gold ions (475 mg/L) solution. And it was shaken at 250 rpm for 24 h at room temperature (298 K). The concentration of the remaining solution was measured using ICP-OES after centrifugation. The precipitate was desorbed with 10% thiourea (2% hydrochloric acid) for 24 h and then the precipitate was washed with pure water for 5 times. Such experiment was performed three times. As shown in Table 7, the adsorption capacity decreased from 860 mg/g to 501.6 mg/g after three times, indicating that TP-AFC can still maintain a large adsorption capacity for gold after three cycles.

### 3.7. Application of TP-AFC in Practical Wastewater

The separation capacity of the adsorbent was tested by laboratory wastewater. The wastewater was firstly adjusted to pH 4 and ICP-OES was used to measure the metal ion concentration in wastewater. Then, 10 mg TP-AFC was added to 20 ml wastewater and oscillated for 24 h, respectively. After centrifugation, the residual concentration of the solution was measured (Figure 7). It can be seen that TP-AFC has higher selectivity for gold ions against the interference of various metal ions. The separation capacity of polyaniline was also tested according to the same procedure. However, polyaniline adsorbed not only gold ion but also cadmium and arsenic ions. Table 8 showed the related parameters of each metal ion on TP-AFC. The distribution coefficient (K_Q_) and selectivity coefficient (K) can be expressed by Equations (10) and (11). The experiment results showed that TP-AFC has a good prospect in practical application.
(12)$$K_{Q}=\frac{Q}{C_{r}}=\frac{C_{i}-C_{r}}{C_{r}}\cdot \frac{V}{m}$$
(13)$$K=\frac{K_{Q\left(Au^{3+}\right)}}{K_{Q\left(coexisting\;ions\right)}}$$

### 3.8. Adsorption Mechanism

The excellent absorption capacity of TP-AFC for gold ion is mainly attributed to the strong interaction between phosphorus-containing functional groups and gold ions. In order to understand the mechanism of gold adsorption by TP-AFC, TP-AFC was characterized by XPS and SEM before and after adsorption. As shown in Figure 8a, the peak of Au4f appeared after adsorption, indicating that gold was absorbed by TP-AFC. As shown in Figure 8b, XPS of gold showed two peaks at 84.1 and 87.59 eV. In addition, XPS of P can be divided into two peaks at 132.78 and 133.71 eV before adsorption (Figure 8c). The peaks shifted to 131.3 and 133.77 eV after adsorption (Figure 8d), indicating that P played an important role in the adsorption process. SEM images of TP-AFC before and after adsorption are shown in Figure 9. It can be clearly seen that the gold is distributed throughout on the adsorbent after adsorption, but the morphology of the adsorbent does not change. In summary, combined with Zeta potential, XPS and SEM, the main mechanism of TP-AFC adsorption of gold is ion exchange and chelation.

## 4. Conclusions

A polyaniline-based adsorbent was successfully prepared for the recovery of gold ions from wastewater. FT-IR, SEM, Zeta potential, and XPS were used to characterize TP-AFC, which has a good adsorption capacity at pH 4. The Langmuir model and pseudo second-order kinetic model describe the adsorption process well. The process is controlled by a single layer chemisorption. Thermodynamic experiments indicated that the adsorption is a spontaneous endothermic process. The adsorption mechanism is mainly the chelation and ion exchange of Au(III) with phosphorus. In the actual wastewater experiment, the coexisting ions have little interference on the recovery of gold. In general, TP-AFC has a good prospect in recycling gold from wastewater.

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
