# Peer review of "Efficient and Selective Adsorption of Gold Ions from Wastewater with Polyaniline Modified by Trimethyl Phosphate: Adsorption Mechanism and Application"

_polymers, 2019, doi:10.3390/polym11040652_

Round 1
Reviewer 1 Report
Please find attached the comments.

Author Response
Dear Editor and Reviewer:
Thank you for your letter and the reviewers’ comments concerning our manuscript entitled “Efficient and selective adsorption of gold ions from wastewater with polyaniline modified by trimethyl phosphate: adsorption mechanism and application”. Those comments are all valuable and very helpful for revising and improving our paper, as well as have important guiding significance for our researches. We have studied the comments carefully and made corrections, and revised portions are marked in red in the revised manuscript. The corrections in the paper and responds to the comments are listed as below:
Reviewer #1: The manuscript entitled “Efficient and selective adsorption of gold ions from wastewater with polyaniline modified by trimethyl phosphate: adsorption mechanism and application” by Wang et al. describe the synthesis of polyaniline based materials for sequestration of gold metals from water. The manuscript needs careful revision of language and grammar, in the present state manscript is not acceptable for publication in “Polymers”. The scientific comments on the manuscript are given below:
Question 1: The experimental section needs significant improvement. The amount in grams for all the materials needs to be converted into moles. The amount in grams has not much scientific significance for a chemical reaction.
Answer: Thanks for your comments, the amount in grams have been converted into moles.
Question 2: For the synthesis of polyaniline, the schematics of polyaniline synthesis do not match with experimental details. In the experimental details, authors mention a second addition of aniline in reaction flask, after the addition of aniline, formaldehyde and HCl in iPrOH. Is this correct? If so why the first addition of aniline was not sufficient for the reaction? Again molar ratios of reactants are required to understand this reaction in details.
Answer: This place is an important mistake and does not require the addition of aniline in the final step. This section has been revised in the manuscript.
Question 3: The description of figure 1 in results and discussion section is wrong. FTIR cannot be expressed in eV. See line 107-111. The units of FTIR are cm-1.
Answer: Thanks for your comment. The mistakes have been modified.
eV → cm-1.
Question 4: For SEM image in figure 2, the authors explain that modification with phosphate does not change the morphology of polyaniline. Why would authors expect a change in morphology by simple adding a functional group to a polymer? FTIR results do shown the presence of phosphate in polyaniline, but what is the degree of modification of amine in polyaniline? Have authors performed an NMR or other amine detecting tests such as Kaiser assay or TNBS assay to find the degree of modification of amine in polyaniline? If so what is the role of amine modification in gold sequestration?
Answer: Analysis of the SEM image in Figure 2 illustrates that the addition of functional groups does not alter the basic morphology of the polyaniline. The results of FTIR confirmed the successful grafting of phosphate.
According to the Suggestions of reviewers, we tried to use ICP and NMR to detect the concentration of phosphorus. The absorbent is insoluble in NMR solvent. The loss of phosphorus volatilization is large in the digestion process. These methods cannot obtain the exact content of phosphorus. Therefore, SEM was used to estimate the grafting rate.
The degree of modification of the amine can be calculated by the data obtained by SEM-EDS and the structural formula of polymer. One nitrogen atom is grafted with two phosphorus atoms. The calculation method is as follows:
CP and CN represent the weight concentration of phosphorus and nitrogen, respectively. MP and MN represent the atomic mass of phosphorus and nitrogen, respectively. The calculation shows that the degree of modification is 20.6%.
Element | Element | Element | Atomic | Weight |
6 | C | Carbon | 57.23 | 46.43 |
7 | N | Nitrogen | 20.86 | 19.74 |
8 | O | Oxygen | 12.61 | 13.64 |
17 | Cl | Chlorine | 0.52 | 0.79 |
15 | P | Phosphorus | 8.78 | 17.81 |
Question 5: The explanation of figure 3 is very vague. What is the net apparent charge of materials at different pH in the absence of gold? How does this change in the presence of gold at different pH? How does polyaniline behave differently than phosphate modified polyaniline?
Answer: The net total surface charge of the sample surface was 2.8. The presence of gold at different pH was measured and presented in the manuscript. Polyaniline showed lower pH tolerance than TP-AFC.
Research of different pH on polyaniline and TP-AFC
Solution pH is an important component of the ability to remove metal ions. The gold solution of pH 2 to 11 was mainly studied on polyaniline and TP-AFC and the concentration was 200 mg/L. Fig. 3(a) showed that the adsorbing capacity varies greatly with the change in pH. Obviously, Polyaniline maintains excellent adsorption capacity in the range of Ph 2 to 4. However, it can be found that the adsorbing capacity of gold ions on TP-AFC is very strong under all acidic conditions (pH 2 to pH 6) and the efficiency of TP-AFC to remove gold ions was optimal at pH 4. When the pH is too high, the hydroxide(OH-) in the solution competes with the adsorbent, resulting in a sharp drop in the adsorbing capacity. As the pH of the solution changes, gold ions mainly exist in the following forms:
The net total surface charge of TP-AFC can be represented by the zero potential (pHPZC). The value of pHPZC charge can be determined by the method described in the literature[35]. The pHPZC of TP-AFC was 2.8. In order to study the adsorption mechanism, the zeta potential of TP-AFC and gold solution were also determined. As shown in Fig. 3 (b), the isoelectric point of TP-AFC was 4.6, indicating that the potential was positive when pH was below 4.6. The gold solution showed a negative charge in the range of pH 2 to pH 4. So, there has an electrostatic effect between TP-AFC and gold.
Question 6: What were the conditions (Ph, salt presence, any organic waste presence) used for kinetics study? How does experimental data obtained for pseudo second order kinetics match with chemistry of the materials? How does the data correlate with polyaniline based materials alone? Are phosphate modified materials any better than polyaniline itself?
Answer: Kinetic studies were carried out at pH 4, ambient temperature (298K), and 200 ppm(gold solution concentration). R2 can be used to prove that the adsorption process conforms to the pseudo second kinetic model. In addition, the value of △q was added to further prove the accuracy. Research of concentration effect on polyaniline and application of polyaniline in practical wastewater were added and indicated that the adsorption capacity and selectivity of polyaniline were inferior to that of TP-AFC.
Question 7: How does experimental data obtained qm value correlated with predicted values of Langmuir isotherm?
Answer: Experimental data obtained qm value correlated with predicted values of Langmuir isothermcan be obtained with △q and have already been added to the manuscript.
Question 8: The quality of SEM image in figure 9b is very poor, and needs to be improved for publication. Gold provide striking phase contrast in SEM or TEM and should be easy to capture than polyaniline alone.
Answer: The image quality of the SEM in Figure 9(b) is indeed very poor. Because of the experiment of adsorbing gold in the manuscript itself, there was no gold spraying before the SEM test because gold spraying will affect the detection of gold ions. In order to verify whether TP-AFC has a stronger adsorption capacity than polyaniline alone, a concentration experiment of polyaniline was added to the manuscript. The experimental results also show that the modified polyaniline has a stronger adsorption capacity.
Question 9: Are these materials recyclable? Recyclability of materials is important for their potential applications in field of water purification.
Answer: Repeatability is important. TP-AFC has also been measured its repeatability. During the repetitive experiment, the adsorption efficiency decreased from 860mg/g to 501.6mg/g after the three cycle. indicating that TP-AFC can still maintain a large adsorption capacity for gold after multiple regenerations.
Reusability and desorption studies of TP-AFC
Recyclability of materials is important for their potential applications in field of water purification. In order to investigate the regenerative capacity of TP-AFC, a repetitive experiment was conducted. 30 mg TP-AFC was added to 60 mL lead ions(475mg/L) solution. And it was shaken at 250 rpm for 24 h at room temperature (298K). The concentration of the remaining solution was measured using ICP after centrifugation. The precipitate was desorbed with 10% thiourea(2% hydrochloric acid) for 24 h and then the precipitate was washed with pure water for 5 times. Such experiment was performed three times. As shown in Table 7, the adsorption capacity decreased from 860mg/g to 501.6mg/g after three times, indicating that TP-AFC can still maintain a large adsorption capacity for gold after multiple regenerations.
Table 7. The regeneration property of TP-AFC
Regeneration times | 1 | 2 | 3 |
Adsorption capacity (mg/g) | 860 | 620 | 501.6 |
Removal rate (%) | 90.5 | 65.3 | 52.8 |

Reviewer 2 Report
The paper presents the efficient and selective adsorption of gold ions from wastewater with polyaniline modified by trimethyl phosphate: adsorption mechanism and application. These issues are important and interesting, but some remarks concerning the preparation of the manuscript should be taken into consideration while revising the paper:
- All references should be adjusted to the Polymers rules,
for example:
J Membrane Sci.; correct: J. Membrane Sci.
React Funct Polym.; correct: React. Funct. Polym.
- All tables, figures and schemes should be described in accordance with the Polymers guidelines,
for example:
Figure 1. FT-IR spectra of AFC and TP-AFC.
- p.2, line 67 , I suggest combining subchapters 2.2.1 and 2.2.2. The TP-AFC synthesis scheme is presented in subchapter 2.2.1. and in 2.2.2 only discussed the synthesis method, it is not logical.
- p. 4, line 121, insert a space between pH and 2 and between pH and 11
- Insert a space between values and units (this applies to the entire text)
- p. 5, line 148, the number 3 is red
- For example, lines 145,146,151,163,164 ,,pseudo’’ should be written with a lowercase letter (this applies to the entire text).
- In Tables 1 and 2 , in the parameters column, complete the units.
- p. 8, line 216 ,,T-model isotherm’’ replace ‘’Temkin-model isotherm’’.
- p. 9, line 224 K change to Ks.
- Table 5 , explain the adsorbents abbreviations under the table.
- p. 10, line 237, insert a space between pH and 4
- p. 10, line, is ln KQ , should be ln KL, analogously in the Figure 6?
- Figure 6, instead of (k-1) it should be( K-1).
- The values of the determined parameters in the tables are given with too great accuracy
Author Response
Dear Editor and Reviewer:
Thank you for your letter and the reviewers’ comments concerning our manuscript entitled “Efficient and selective adsorption of gold ions from wastewater with polyaniline modified by trimethyl phosphate: adsorption mechanism and application”. Those comments are all valuable and very helpful for revising and improving our paper, as well as have important guiding significance for our researches. We have studied the comments carefully and made corrections, and revised portions are marked in red in the revised manuscript. The corrections in the paper and responds to the comments are listed as below:
Reviewer #2: The paper presents the efficient and selective adsorption of gold ions from wastewater with polyaniline modified by trimethyl phosphate: adsorption mechanism and application. These issues are important and interesting, but some remarks concerning the preparation of the manuscript should be taken into consideration while revising the paper:
Question 1: All references should be adjusted to the Polymers rules, for example:
J Membrane Sci.; correct: J. Membrane Sci. React Funct Polym.; correct: React. Funct. Polym.
Answer: All references have been adjusted to the Polymersrules.
Question 2: All tables, figures and schemes should be described in accordance with the Polymers guidelines, for example:
Figure 1. FT-IR spectra of AFC and TP-AFC.
Answer: All tables, figures and schemes have been described in accordance with the Polymers guidelines.
Question 3: p.2, line 67 , I suggest combining subchapters 2.2.1 and 2.2.2. The TP-AFC synthesis scheme is presented in subchapter 2.2.1. and in 2.2.2 only discussed the synthesis method, it is not logical.
Answer: This two parts have been presented together.
Question 4: p. 4, line 121, insert a space between pH and 2 and between pH and 11. Insert a space between values and units (this applies to the entire text) p. 5, line 148, the number 3 is red
Answer: All the mistakes has been modified.
Question 5: For example, lines 145,146,151,163,164 ,,pseudo’’ should be written with a lowercase letter (this applies to the entire text).
Answer: Those part has been written with a lowercase letter.
Question 6: In Tables 1 and 2 , in the parameters column, complete the units.
Answer: The units were added to Table 1 and 2.
Question 7: p. 8, line 216 ,,T-model isotherm’’ replace ‘’Temkin-model isotherm’’.
Answer: T-model isotherm’’ has replaced ‘’Temkin-model isotherm’’.
Question 8: p. 9, line 224 K change to Ks.
Answer: This place K has been changed to Ks.
Question 9: Table 5 , explain the adsorbents abbreviations under the table.
Answer: The adsorbents abbreviations were explained in the table.
Table 5. Comparison of the adsorbing capacity for the gold adsorption
adsorbents | (mg/g) | pH | litertures | |||
APS-LCP (lignocellulosic) | 261.36 | 4.0 | [30] | |||
PS-APD resin (resin) | 278.5 | 4.0 | [31] | |||
n-AMPRs (cellulose) | 537 | 2.0 | [32] | |||
D301-g-THIOPGMA (resin) | 326 | 2.0 | [33] | |||
BHJC (buckwheat hulls) | 425.5 | 2.5 | [34] | |||
TP-AFC (polyaniline- trimethyl phosphate) | 881 | 4 | This work | |||
Question 10: p. 10, line 237, insert a space between pH and 4
Answer: A space was added to pH and 4.
Question 11: p. 10, line, is ln KQ , should be ln KL, analogously in the Figure 6?
Answer: ln KQ is not ln KL. KQ is the distribution coefficient and KL is the constant of Langmuir model.
Question 12: Figure 6, instead of (k-1) it should be( K-1).
Answer: (k-1) has been corrected as ( K-1).
Question 13: The values of the determined parameters in the tables are given with too great accuracy
Answer: All values in the tables have been modified with appropriate precision.

Round 2
Reviewer 1 Report
The authors have provided detailed answers to queries, I suggest publishing this manuscript after some minor revisions.
revision of language and grammar
Authors have provided detailed answers to queries but they are not well-discussed in the manuscript. I suggest the authors discuss the answers of question 4 & 5 in the manuscript as well.
Author Response
Dear Editor and Reviewer:
Thank you for your letter and the reviewers’ comments concerning our manuscript entitled “Efficient and selective adsorption of gold ions from wastewater with polyaniline modified by trimethyl phosphate: adsorption mechanism and application”. Those comments are all valuable and very helpful for revising and improving our paper, as well as have important guiding significance for our researches. We have studied the comments carefully and made corrections, and revised portions are marked in red in the revised manuscript. The corrections in the paper and responds to the comments are listed as below:
1 revision of language and grammar
Answer: Thanks for your comments, we have carefully corrected the language and grammar.
2 Authors have provided detailed answers to queries but they are not well-discussed in the manuscript. I suggest the authors discuss the answers of question 4 & 5 in the manuscript as well.
Answer: the answers of question 4 were added to Page 5 and Figure 2 (c). The answers of question 5 were added to Page 5 to Page 6.
